# TimeRewarder: Learning Dense Reward from Passive Videos via Frame-wise Temporal Distance

## Abstract

Designing dense rewards is crucial for reinforcement learning (RL), yet in robotics it often demands extensive manual effort and lacks scalability. One promising solution is to view *task progress* as a dense reward signal, as it quantifies the degree to which actions advance the system toward task completion over time. We present *TimeRewarder*, a simple yet effective reward learning method that derives progress estimation signals from passive videos, including robot demonstrations and human videos, by modeling temporal distances between frame pairs. We then demonstrate how *TimeRewarder* can supply step-wise proxy rewards to guide reinforcement learning. In our comprehensive experiments on ten challenging Meta-World tasks, we show that *TimeRewarder* dramatically improves RL for sparse-reward tasks, achieving nearly perfect success in 9/10 tasks with only 200,000 interactions per task with the environment. This approach outperformed previous methods and even the manually designed environment dense reward on both the final success rate and sample efficiency. Moreover, we show that *TimeRewarder* can exploit real-world human videos, highlighting its potential as a scalable approach path to rich reward signals from diverse video sources.

## 1 Introduction

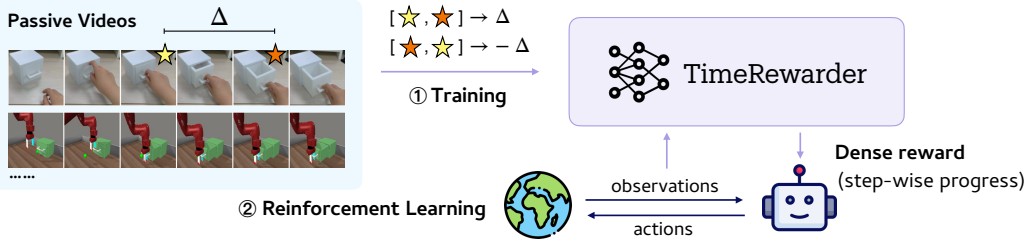

Figure 1: Overview of *TimeRewarder*. Mirroring how humans infer task progression by observing others, *TimeRewarder* distills frame-wise temporal distances from expert videos and converts them into dense reward signals, thereby enabling reinforcement learning free of manually engineered rewards or action annotations.

Reinforcement learning (RL) has long served as a principal paradigm for robotic skill acquisition (Ibarz et al., 2021; Tang et al., 2025). Yet, many of its most notable successes so far rely highly on carefully designed reward functions that are dense and task-instructive (Cheng et al., 2024; Nai et al., 2025). Designing such high-quality rewards remains labor-intensive, as they often require significant domain expertise, extensive hyperparameter tuning, or privileged access to ground-truth environments, especially for robotic manipulations (Ng et al., 1999a; Levine et al., 2016; Rajeswaran et al., 2017; Roy et al., 2021). These challenges incurred during manual reward design severely constrain the scalability of RL approaches, motivating the development of automated reward learning mechanisms that can alleviate human effort.

Dense reward function design for robotics often exploits explicit prior knowledge of the task's typical progression, which estimates the distance between the current state and task completion, as

well as assesses whether the current action contributes to efficient task accomplishment (Todorov, 2004; Levine et al., 2016; Silver et al., 2021). Expert demonstrations provide a natural source of this progression knowledge: the temporal ordering of video frames directly reflects task advancement. Importantly, such signals can be derived even from passive videos, which are easy to obtain and require neither action annotations nor privileged supervision. As a result, automatic reward learning from passive videos can significantly expand the scalability of RL.

Building on this idea, we introduce *TimeRewarder* (Figure 1), which comprehends how the task proceeds by learning to predict *temporal distances* between arbitrary frames from action-free expert demonstrations. The temporal distance reflects the *task progress* between two frames: which frame is closer to task completion and by how much. When turning to the RL exploration phase, the predicted progress distances between adjacent frames can naturally serve as dense reward signals. The step-wise reward quantifies exactly how much the agent is advancing or regressing at each moment, guiding the agent toward accomplishing the task by implicitly imitating the expert demonstrations.

We evaluate *TimeRewarder* in the imitation-from-observation setting, where only expert videos are available and no expert action labels or dense environment rewards are provided. On 10 Meta-World (Yu et al., 2020) manipulation tasks with 100 demonstrations per task, *TimeRewarder* surpasses all baselines on 9 tasks in both success rate and sample efficiency. This performance gain highlights the high quality of the reward produced by *TimeRewarder*: it effectively assigns credits to partial progress and penalizes unproductive behaviors even on out-of-distribution transitions along the agent trajectories, thus providing strong instructive guidance to the RL process.

## 2 RELATED WORKS

Previous work has explored methods of learning from observation-only demonstrations, providing agents with task-relevant supervision when environmental rewards are sparse or inaccessible.

**Action recovery.** Model-based approaches (Nair et al., 2017; Torabi et al., 2018a; Pathak et al., 2018; Edwards et al., 2019; Radosavovic et al., 2021; Fan et al., 2022; Baker et al., 2022; Liu et al., 2022; Ramos et al., 2023) attempt to recover missing actions in expert demonstrations by training inverse dynamics models from online exploration data, and apply behavioral cloning on annotated videos. However, these methods necessitate the collection of vast amounts of transition data to train reliable action-recovery models, and this training must be performed iteratively online to ensure the state distribution of the expert demonstrations is adequately covered. Such a delicate and unstable process limits their practical deployment in potential real-world robotic scenarios.

**Inverse RL.** Instead of explicitly recovering actions for behavior cloning, Inverse RL aims to build reward functions from expert demonstrations (and online interactions if needed) to guide policy updates within a standard RL paradigm. Trajectory-matching methods (Dadashi et al., 2020; Yang et al., 2019; Jaegle et al., 2021; Chen et al., 2021; Haldar et al., 2023; Liu et al., 2024) measure rollout–expert similarity as a reward signal, while adversarial imitation learning (Ho & Ermon, 2016; Torabi et al., 2018b; Cai et al., 2019) trains a discriminator to distinguish agent from expert transitions. With the advance of generative models, some recent works (Escontrela et al., 2023; Huang et al., 2024) train video generation models and take the likelihood of rollout frames produced by this model as the reward. Despite the progress of these methods, they face challenges such as high online computational cost (Haldar et al., 2023; Escontrela et al., 2023), training instability (Ho & Ermon, 2016), or reward hacking (Escontrela et al., 2023).

**Progress-based reward learning.** Within inverse RL, some methods define proxy rewards by exploiting the temporal structure of demonstrations, where the ordering of frames along a trajectory provides an implicit measure of *task progress*. TCC (Dwibedi et al., 2019) enforces cycle-consistency in time for correspondence, while Arrow of Time (Wei et al., 2018) exploits temporal irreversibility for representation learning. TCN (Sermanet et al., 2018) pulls temporally adjacent frames together in the latent visual representation space while pushing distant ones apart. However, as pointed out by Ma et al. (2022), standard TCN enforces only coarse temporal consistency and produces non-locally smooth representations. Building on this, VIP (Ma et al., 2022) estimates frame–goal distances using implicit time-contrastive learning. However, we found this objective difficult to optimize reliably. GVL (Ma et al., 2024) uses vision-language models to infer temporal orders from shuffled frames, yet we observed that the outputs of these large models can be

inconsistent, limiting their effectiveness in building reward functions. Rank2Reward (Yang et al., 2024) learns to predict the temporal order of adjacent frame pairs, providing lightweight local rewards; PROGRESSOR (Ayalew et al., 2024) considers triples of frames to estimate the relative position of an intermediate frame between start and goal states. However, both Rank2Reward and PROGRESSOR report that rewards trained solely on expert data tend to overestimate progress for out-of-distribution states, necessitating online refinement for stable policy learning.

In contrast, our method directly estimates *frame-wise temporal distances*, producing more accurate and stable proxy rewards. Once trained on expert videos, the reward model can be frozen during RL, eliminating the need for online updates. These properties enable its potential scaling to large and diverse demonstration datasets, making it better suited for practical policy learning.

## 3 PRELIMINARIES

### 3.1 LEARNING FROM ACTION-FREE DEMONSTRATIONS

We study the problem of learning policies from action-free expert demonstrations. Specifically, the agent has access to a dataset of expert RGB videos besides an environment to interact with. We resolve the problem from the RL perspective, by deriving a proxy reward from the action-free demonstrations, which is used to guide downstream policy optimization.

Formally, we consider an agent interacting with a finite-horizon Markov Decision Process $(\mathcal{S}, \mathcal{A}, \mathcal{P}, \mathcal{R}, \gamma, T)$, where $\mathcal{S}$ is the state space, $\mathcal{A}$ the action space, $\mathcal{P}$ the transition dynamics, $\mathcal{R}$ the reward function, $\gamma$ the discount factor, and $T$ the horizon. We assume the agent can not access states $s_t \in \mathcal{S}$ directly, but only high-dimensional visual observations $o_t \in \mathcal{O}$ in the form of RGB images. Moreover, the environmental reward function $\mathcal{R}$ provides only sparse binary success signals indicating whether the task is completed or not, which is easily obtainable via human annotation or vision-language model API.

Such sparse signals are far from enough for guiding efficient policy optimization. To overcome this, we derive a proxy reward from the expert data, hoping that the agent can receive instructive learning signals even when the environmental reward remains zero during exploration. We denote the expert dataset as $D^e = \{\tau_i^e\}$, where $\tau^e = (o_1^e, o_2^e, \ldots, o_T^e)$ represents observation trajectories. The goal is to recover a proxy reward function $\hat{\mathcal{R}}$ from $D^e$, such that a policy $\pi^{\hat{\mathcal{R}}}$ trained on this reward:

$$\pi^{\hat{\mathcal{R}}} = \arg\max_{\pi} \mathbb{E}\left[\sum_{t=1}^{T} \gamma^{t-1}\hat{\mathcal{R}}(o_t, o_{t+1})\right] \tag{1}$$

can successfully accomplish the task.

### 3.2 PROGRESS-BASED REWARD DESIGN

Since the agent's ultimate objective is to reach a goal state, the distance to task completion can be interpreted as a measure of task progress, which can inform reward design. This idea is closely related to potential-based reward shaping (Ng et al., 1999b), where the reward at each transition is defined as the change in a potential function $V(o)$ that measures progress from $o$ toward the goal:

$$r_t = \hat{\mathcal{R}}(o_t, o_{t+1}) = V(o_t) - \gamma V(o_{t+1}). \tag{2}$$

Such progress-based proxy rewards offer two primary benefits: (1) *Generality*: Task progress is a high-level signal that is implicitly encoded in expert demonstrations, avoiding the need for hand-crafted reward design. (2) *Action-free learning*: Progress can be inferred directly from passive video data, without requiring access to action labels. These properties yield dense and temporally consistent feedback, enabling policy learning from action-free video demonstrations.

## 4 METHOD

We introduce *TimeRewarder*, a framework that derives dense proxy rewards for downstream RL by estimating task progress from action-free expert videos. The central idea is to model progress as

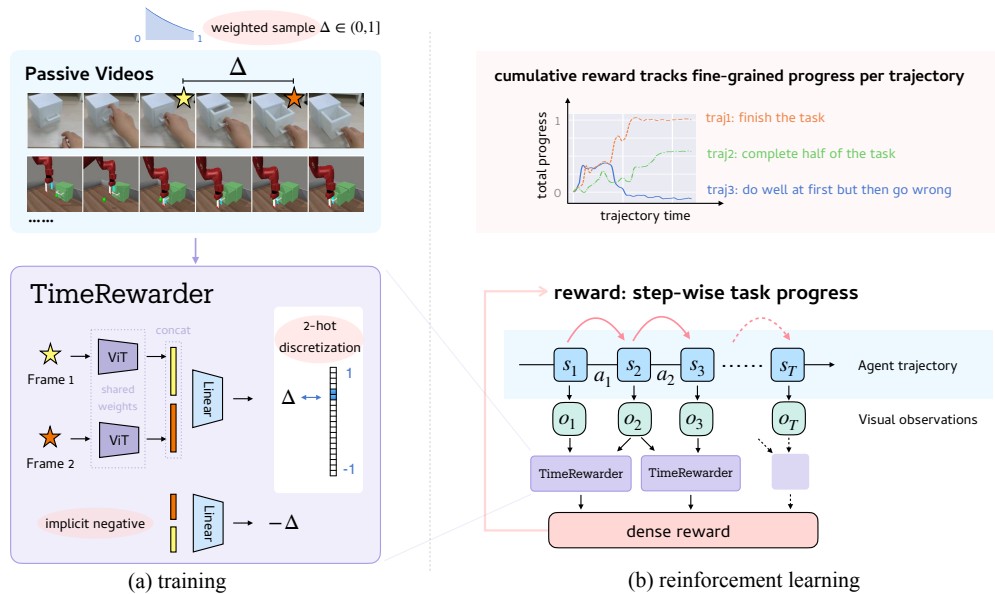

(a) training

(b) reinforcement learning

Figure 2: *TimeRewarder* framework. *TimeRewarder* learns step-wise dense rewards from passive videos by modeling intrinsic temporal distances, enabling robust progress scoring that assigns high values to states reflecting task advancement, while penalizing suboptimal actions lacking meaningful contribution to task progression, thereby facilitating effective policy learning.

a *temporal distance prediction problem*: learning to estimate the temporal distance between two observations in a trajectory. In this section, we (1) formalize the construction and training of *TimeRewarder*, (2) present its application in deriving reward functions for RL, and (3) provide a theoretical justification demonstrating that temporal distance aligns naturally with task progress.

## 4.1 TRAINING WITH FRAME-WISE TEMPORAL DISTANCE

We train *TimeRewarder*, a progress model $F_\theta : \mathcal{O} \times \mathcal{O} \to \mathbb{R}$, on expert demonstrations $D^e$. The model learns to predict the normalized temporal distance between two ordered frames $(o_u^e, o_v^e)$, providing a dense signal of task progress. As shown in Figure 2 (a), given two frames $(o_u^e, o_v^e)$ from an expert trajectory, their normalized temporal distance is computed as:

$$d_{uv} = \frac{v - u}{T - 1} \in [-1, 1], 1 \le u, v \le T, \tag{3}$$

so that $F_\theta$ is trained in a self-supervised manner, taking two ordered frames and predicting the relative temporal distance between them.

To be effective as a reward signal, $F_\theta$ must satisfy two key principles: (1) **Suboptimality Awareness** — generalize beyond expert data and assign lower scores to suboptimal behaviors which are unseen in $D^e$; (2) **Fine-grained Temporal Resolution** — capture fine-grained progress, particularly between adjacent steps.

For *suboptimal awareness*, *TimeRewarder* naturally realize **Implicit Negative Sampling**: the frame indices $u$ and $v$ in equation 3 can appear in either forward or backward order, so the normalized temporal distance $d_{uv}$ ranges from $-1$ to $1$. A positive value indicates forward progression toward the goal, while a negative value indicates backward progression, naturally corresponding to movement away from task completion, simulating suboptimal or incorrect behaviors. This formulation imposes an antisymmetric structure on the learning objective, thereby discouraging trivial memorization shortcuts (Ma et al., 2024).

As for *fine-grained temporal resolution*, we aim to enhance the model's ability to recognize progress at the step level, i.e., between adjacent frames, so that the learned metric can provide reliable step-wise rewards. To this end, we introduce **Exponentially Weighted Pair Sampling**: the temporal

interval $\Delta = |v - u|$ in a frame pair $(o_u^e, o_v^e)$ is sampled according to

$$P(\Delta) \propto \exp(-\lambda\Delta), \quad \Delta \in \{1, \ldots, T - 1\}, \tag{4}$$

where $\lambda > 0$ controls the bias toward shorter intervals while still ensuring coverage of longer horizons. This sampling scheme emphasizes fine-grained local differences while retaining the ability to capture broader temporal dependencies.

Besides, to ensure numerical stability and maintain accuracy during the optimization process, we employ **Two-hot Discretization** (Wang et al., 2024) to discretize the scalar temporal distance $d_{uv} \in [-1, 1]$. Specifically, the target range $[-1, 1]$ is uniformly partitioned into $K$ bins (we set $K = 20$ by default). For a given $d_{uv}$, we compute a soft two-hot distribution $\mathbf{y}_{uv} = \Phi(d_{uv}) \in \mathbb{R}^K$ that assigns non-zero mass only to the two nearest bins. The progress model $F_\theta$ outputs a logit vector $\hat{\mathbf{y}}_{uv} = F_\theta(o_u^e, o_v^e) \in \mathbb{R}^K$, and the training objective is the cross-entropy loss:

$$\min_\theta \ \mathbb{E}\big[ - \mathbf{y}_{uv}^\top \log \mathrm{softmax}(\hat{\mathbf{y}}_{uv})\big]. \tag{5}$$

Through this training, $F_\theta$ learns a robust notion of temporal progress inside any ordered frame pairs from purely observational passive video data.

## 4.2 POLICY LEARNING WITH TEMPORAL DISTANCE REWARD

Then, we utilize $F_\theta$ to provide dense proxy rewards for RL. As illustrated in Fig. 2 (b), for each policy rollout, *TimeRewarder* computes adjacent frame distances as step-wise rewards:

$$r_{\mathrm{TR}}(o_t, o_{t+1}) = \hat{d}_{t,t+1} = \Phi^{-1}\big[F_\theta(o_t, o_{t+1})\big] \in [-1, 1], \tag{6}$$

where the output logits of $F_\theta$ have been converted back to a scalar value.

During policy optimization, we combine this progress-based dense reward with a sparse success signal:

$$r_t = r_{\mathrm{TR}}(o_t, o_{t+1}) + \alpha \cdot r_{\mathrm{success}}(o_t), \tag{7}$$

where $r_{\mathrm{success}} : \mathcal{O} \to \{0, 1\}$ is a binary success indicator (1 if successful, 0 otherwise), and $\alpha \geq 0$ is a weight constant.

Although $F_\theta$ is trained solely on expert trajectories, its design ensures natural generalization to diverse behaviors. Suboptimal behaviors—such as stalls, loops, or regressions—receive lower or even negative rewards, while meaningful partial progress is still recognized and positively rewarded. This graded, step-wise feedback provides informative signals for exploration, guiding the agent to recover from failures and make constructive progress toward task completion. Together with the sparse success signal, this mechanism allows *TimeRewarder* to produce dense and informative rewards throughout training, which underlies its empirical effectiveness demonstrated in Section 5.

## 4.3 THEORETICAL JUSTIFICATION

We provide a theoretical justification for our motivation that the task progress in expert videos can be formalized in terms of temporal distance. For each expert trajectory $\tau^e = (o_1^e, \ldots, o_T^e)$, we treat the final frame as the goal observation $o_g^e$. If the true reward is absent, the normalized step cost can be approximated as $\frac{1}{T-1}$, where $T$ is the trajectory length. The progress (potential) of each observation $o_t^e$ can thus be expressed as:

$$V(o_t^e) = \mathbb{E}\left[\sum_{k=t}^{T-1} \frac{1}{T-1}\gamma^{k-t}\mathbf{1}\{o_k^e \neq o_g^e\}\right], \quad V(o_g^e) = 0, \tag{8}$$

where $\mathbf{1}\{\cdot\}$ denotes the indicator function.

Under the assumption of expert optimality, this potential satisfies the Bellman equation.

$$V(o_t^e) = \mathbb{E}\left[\frac{1}{T-1} + \gamma V(o_{t+1}^e)\right]. \tag{9}$$

Generally, $\gamma$ is set to a large value close to 1, so along the expert trajectory, the progress reward in equation 2 approximates the per-step temporal distance $\frac{1}{T-1}$. This highlights that frame-wise temporal distance provides a natural and theoretically motivated measure of progress.

## 5 EXPERIMENTS

In this section, we assess the performance of *TimeRewarder*. We present the experiment setup, evaluate *TimeRewarder* against baselines, and do ablation studies of its key components.

### 5.1 EXPERIMENT SETUP

**Evaluation Benchmark.** We evaluate *TimeRewarder* and other methods on ten challenging Meta-World (Yu et al., 2020) manipulation tasks (see Appendix A.1 for details). For each task, we provide 100 action-free expert videos generated by Meta-World's scripted policies. For three tasks among them, we further consider a cross-domain setting where only one in-domain expert video is provided per task, supplemented with 20 real-world human demonstration videos.

**Implementation Details.** We use a CLIP-pretrained ViT-B (Radford et al., 2021; Dosovitskiy et al., 2021) as the visual backbone of *TimeRewarder*. During training, frame pairs are independently encoded, concatenated, and passed through a linear layer to predict discretized temporal distances. Both the ViT-B encoder and linear layer are trainable. For RL, *TimeRewarder* is integrated with DrQ-v2 (Yarats et al., 2021), and the whole network is frozen, providing dense step-wise rewards from adjacent observation frames. See Appendix A.4.3 for hyperparameters.

**Baselines.** We compare TimeRewarder against eight baselines, grouped into three categories:

1. *Progress-Based Reward Learning:* **PROGRESSOR** (Ayalew et al., 2024) fits a Gaussion model to estimate relative frame positions between initial and goal as rewards; **Rank2Reward** (Yang et al., 2024) estimates temporal rank between frames as rewards; and **VIP** (Ma et al., 2022) trains an implicit value model to estimate task progress of each frame. For fair comparison, following their settings, goal frames sampled from expert videos are provided to PROGRESSOR and VIP.

2. *Imitation Learning from Observations:* **GAIfO** (Torabi et al., 2018b), **OT** (Papagiannis & Li, 2022), and **ADS** (Liu et al., 2024) compute rewards online by comparing rollouts to expert videos. GAIfO uses a discriminator, OT applies Wasserstein distance via Optimal Transport (Villani et al., 2009), and ADS extends OT with curriculum scheduling on the discount factor to better handle progress-dependent tasks.

3. *Privileged Methods:* For reference, we also report results of policies with access to privileged information: **BC** (Bain & Sammut, 1995) trains a behavior cloning policy with expert actions, and **Environment reward** uses Meta-World's ground-truth dense reward.

For the seven baselines involving reinforcement learning (except **BC**), we uniformly adopt DrQ-v2 (Yarats et al., 2021) as the underlying RL algorithm for fair comparison.

### 5.2 PERFORMANCE OF TIMEREWARDER

We address the following four questions to structure our experimental results and analysis, to demonstrate the superior performance achieved by *TimeRewarder* against the baselines.

**Question 1.** *Does TimeRewarder provide correct task progress for unseen success trajectories rather than relying on memorization?*

A well-shaped reward should encourage successful rollouts with monotonic progress, even when trajectories differ from training demonstrations in object positions or motion paths. We test *TimeRewarder* and progress-based reward baselines under the Value-Order Correlation (VOC) metric (Ma et al., 2024), which evaluates the alignment between predicted values and temporal order ($+1$ for perfect monotonicity increasing, $0$ for no correlation, $-1$ for inverse). Specifically, we train *TimeRewarder* and VIP on 100 expert demonstrations and test them on 100 held-out expert videos. To further strengthen the empirical comparison, we introduce GVL (Ma et al., 2024) implemented with `Gemini-1.5-Pro` (Team et al., 2024) as an additional baseline, where we follow its few-shot setting by giving 5 expert videos as context and another 5 for testing, where 32 frames are uniformly sampled from each video. Rank2Reward and PROGRESSOR are excluded due to a lack of value functions indicating progress. As shown in Figure 3, *TimeRewarder* consistently achieves the highest VOC scores, confirming its strong temporal coherence and generalization to unseen trajectories.

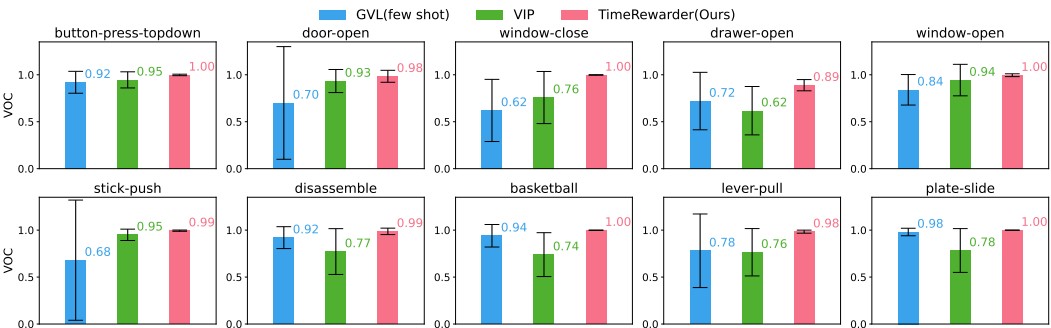

Figure 3: Value–Order Correlation (VOC) on held-out expert videos. Higher is better.

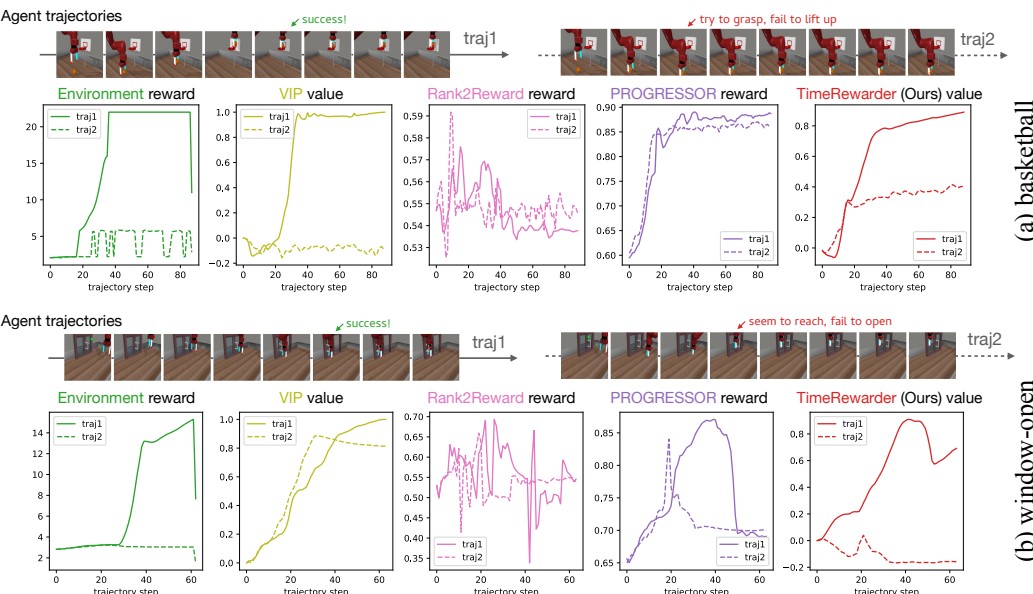

Figure 4: Reward/value curves on successful (`traj1`) vs. failed (`traj2`) rollouts for two tasks. *TimeRewarder* and VIP output *values* (cumulative progress), PROGRESSOR outputs step-wise *rewards*, while Rank2Reward is visualized through its pairwise ordering reward signals. *TimeRewarder* provides the most informative and temporally coherent feedback.

**Question 2.** *Can TimeRewarder identify suboptimal behavior in rollout trajectories?*

Reward models trained on demonstrations of only successful behaviors inevitably face out-of-distribution transitions during RL exploration, where they may misinterpret them by either overestimating failures or undervaluing successes. We select one representative successful (`traj1`) and one failed (`traj12`) trajectory from two tasks, and visualize the progress estimates of *TimeRewarder* against three baselines in Figure 4.

In the *basketball* task, where `traj2` grasps but never lifts the ball, VIP ignores partial progress and PROGRESSOR saturates after grasping, while *TimeRewarder* cleanly captures half-success and then separates completion from failure. In the *window-open* task, where `traj2` mimics opening motions midair without contacting the handle, VIP is misled by visual similarity and PROGRESSOR gives spurious early spikes which can mislead exploration, whereas *TimeRewarder* increases values only upon meaningful interaction. Rank2Reward, limited to pairwise orderings, fails to produce consistent distinctions. These comparative results demonstrate *TimeRewarder*'s unique capacity for temporally coherent and causally grounded feedback under distribution shift—significantly outperforming previous methods in distinguishing productive from unproductive behaviors.

**Question 3.** *Can TimeRewarder improve reinforcement learning performance?*

We present the downstream RL performance of *TimeRewarder* against baselines in Figure 5. Specifically, we implement DrQ-v2 with rewards summed up from the proxy rewards produced by

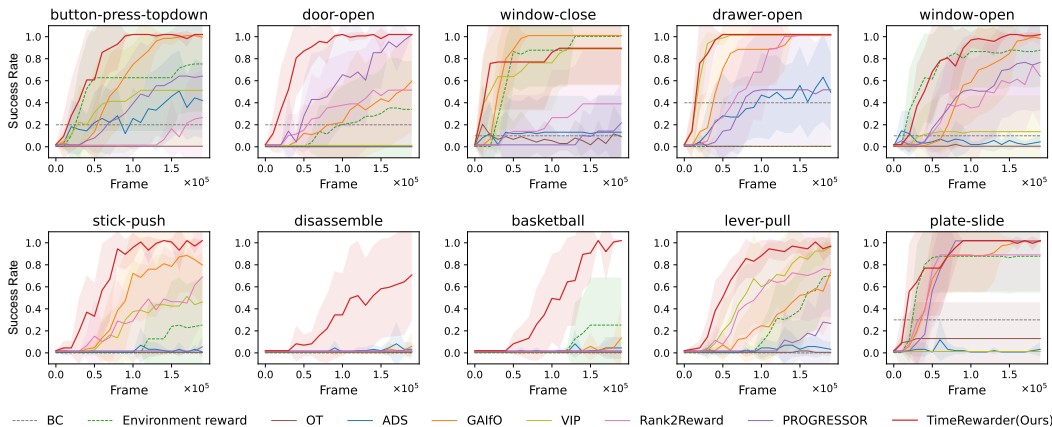

Figure 5: Performance of reinforcement learning with sparse environment success signals and dense proxy rewards from each method. Curves show mean ± s.d. over eight seeds. Dashed lines indicate reference settings of behavior cloning (BC) and environment dense reward supervision.

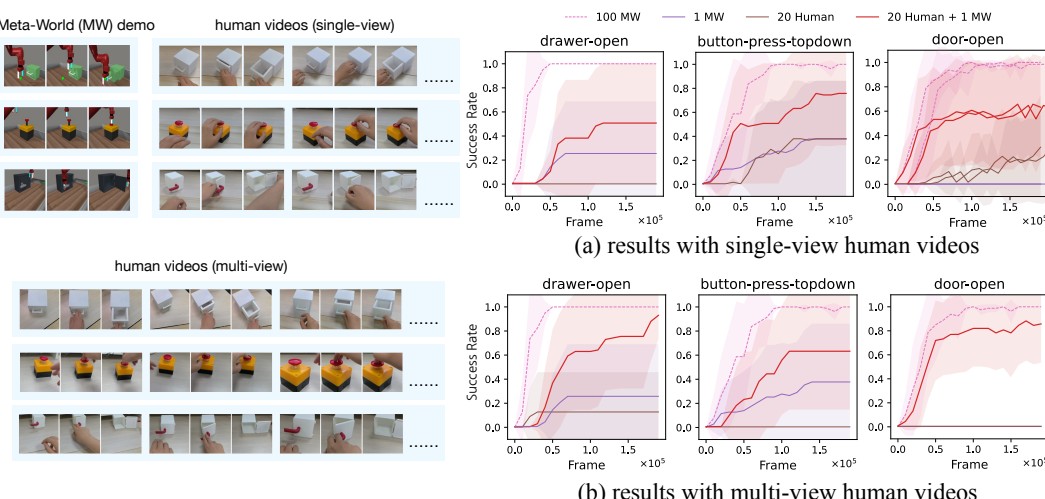

(a) results with single-view human videos

(b) results with multi-view human videos

Figure 6: Cross-domain reward learning. *TimeRewarder* improves performance by leveraging 20 unlabeled human videos alongside only 1 in-domain Meta-World demonstration per task, demonstrating its ability to utilize cross-domain visual data. Curves show mean ± s.d. over eight seeds.

these methods and the environmental binary success signals, similar to equation 7. We see that *TimeRewarder* attains the highest final success rate and the greatest sample efficiency on 9 of 10 tasks. Remarkably, *TimeRewarder* also outperforms policies trained with dense **Environment reward** on 9 tasks, which is commonly treated as an upper bound. These results demonstrate that *TimeRewarder*, a progress-based reward learning method, not only eliminates the need for manual reward design but can even surpass it in effectiveness. Additional experiments without environment success signals are provided in Appendix A.3.2.

**Question 4.** *Can TimeRewarder generalize across different domains and even embodiments?*

To test cross-domain generalization, we choose 3 tasks and build their corresponding copies in real-world. For these 3 tasks we collect 20 human demonstrations individually under each of the following 2 camera settings: fixed viewpoint or varying viewpoints. Such cross-domain videos, together with a single in-domain expert video from the original Meta-World environment, are then provided to *TimeRewarder* for reward learning and downstream RL. As shown in Figure 6, training on either human-only (brown) or Meta-World-only (purple) data yields low success rates, but combining them (red) substantially improves performance. These results highlight the ability of *TimeRewarder* to leverage cross-domain, unlabeled video data for reward learning, even when in-domain supervision is scarce. The full set of human videos is shown in Appendix A.2.

## 5.3 ABLATION STUDIES

In Figure 7, we evaluate the contribution of each methodological component in Section 4.1 through controlled removals:

**Effect of Implicit Negative Sampling:** Implicit negative sampling enforces *suboptimal awareness* by treating reverse-ordered frame pairs as implicit negatives, simulating failures during training. Removing it and predicting only forward progress $\in [0, 1]$ causes sharp drops in *stick-push* and *basketball* (orange line), where failed grasps must be penalized. Without negatives, the model overestimates such failures as partial success. PROGRESSOR, which also lacks this mechanism, similarly collapses (Figure 5), highlighting its necessity.

**Effect of Weighted Sampling:** Weighted sampling enforces *fine-grained temporal resolution* by emphasizing short frame intervals while still covering long horizons. Replacing it with uniform sampling reduces performance in *stick-push* and *window-open* (pink line), where precise interactions are required. Without focusing on adjacent frames, the model misses subtle cues, yielding ambiguous rewards that fail to guide effective, precise action learning.

**Effect of Discretization:** Two-hot discretization ensures *numerical stability* and sharp progress boundaries by binning temporal distances. Replacing it with direct regression causes large drops in *basketball* and *disassemble* (purple line), where long setup phases are followed by brief decisive actions (e.g., lifting the ball or ring). Direct regression smooths over these moments, failing to distinguish success from near-success, while discretization preserves sharp transitions and provides stronger completion incentives.

We also evaluate three alternative designs (details in Appendix A.4.1): (1) **only from init** measures progress only relative to the initial frame; (2) **single frame input** predicts the progress of each single frame instead of relative progress between two frames; and (3) **order prediction** is inspired by GVL (Ma et al., 2024) and reconstructs sequences from shuffled frames. All perform worse: the former two settings lacks temporal expressiveness, while the third one adds complexity without benefit, underscoring the effectiveness of *TimeRewarder*.

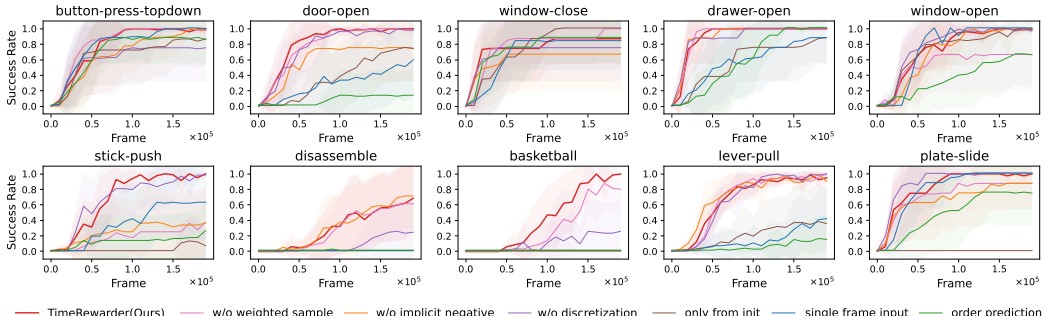

Figure 7: Ablation study results. Curves show mean ± s.d. over eight seeds.

## 6 CONCLUSION

We present *TimeRewarder*, a simple yet effective method that produces dense instructive rewards by learning to predict temporal distances from action-free expert videos. This approach captures fine-grained task progress, naturally accounts for suboptimal behaviors, and provides informative step-wise feedback for RL. Experiments on diverse robotic manipulation tasks demonstrate that *TimeRewarder* not only outperforms prior reward learning methods but also surpasses environment-supplied dense rewards, in terms of both success rate and sample efficiency. Besides, *TimeRewarder* showed successful cross-domain learning ability by leveraging real-world human videos to improve policy learning, when in-domain data is limited.

In a word, *TimeRewarder* provides a promising direction for reducing reliance on manual reward engineering. Although current limitations emerge on tasks with frequent back-and-forth motions, we expect them to be addressed by future hierarchical or memory-augmented progress models, so that scalable "watch-to-act" skill acquisition from in-the-wild video becomes truly attainable.

## REPRODUCIBLE STATEMENT

With the code attached in our supplementary materials and the hyperparameter settings in Appendix A.4.3, the experiment results are highly reproducible. We also utilize sufficient random seeds in Section 5 to ensure reproducibility.

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

# A APPENDIX

## A.1 TASKS FOR EVALUATION

In this paper, we experiment with the following 10 tasks from the Meta-World suite (Yu et al., 2020):

1. **Button press topdown:** to press a button from the top.

2. **Door open:** to open a cabinet door with a handle.

3. **Window close:** to close a sliding window with a handle.

4. **Drawer open:** to open a cabinet drawer with a handle.

5. **Window open:** to open a sliding window with a handle.

6. **Stick push:** to pick up a stick and push a kettle with the stick.

7. **Disassemble:** to pick and remove a nut from a peg.

8. **Basketball:** to pick up a basketball and dump it into a basket.

9. **Lever pull:** to pull a lever up 90 degrees.

10. **Plate slide:** to push a plate into the goal area.

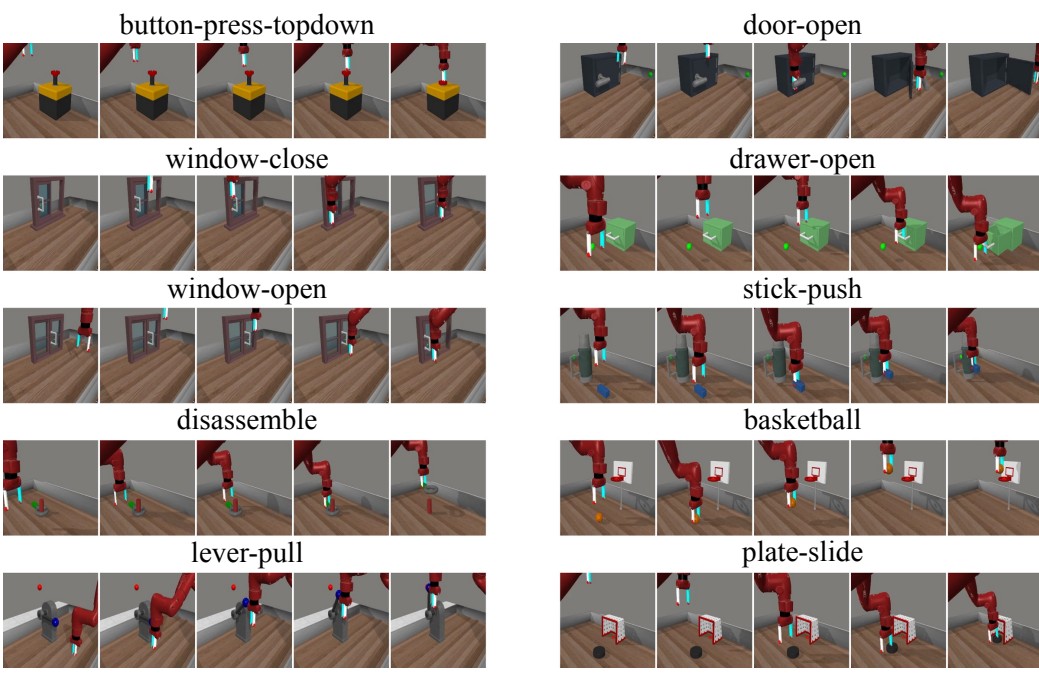

Figure 8: Meta-World tasks used in our paper.

## A.2 HUMAN VIDEO DATASETS FOR CROSS-DOMAIN EXPERIMENTS

This section presents the complete set of human videos used in the cross-domain experiments across three tasks. Each task includes 20 videos recorded in *single-view* (fixed viewpoint) and 20 videos recorded in *multi-view* (varying viewpoints) conditions. These videos differ from the robot setting in embodiment and background, and contain no action or state annotations.

The full set of videos for each task in both conditions is shown in Figure 9 and Figure 10.

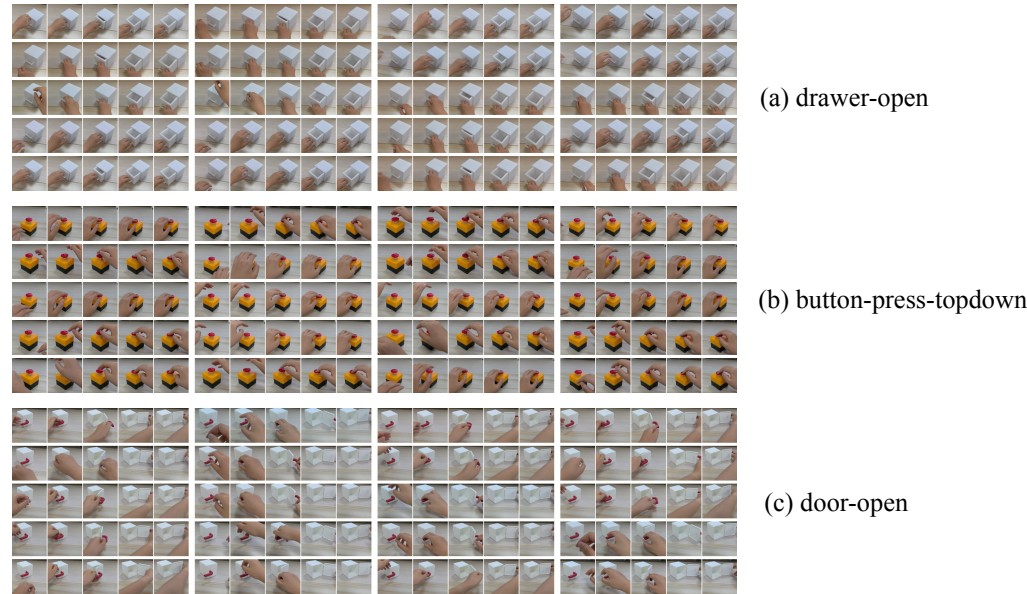

Figure 9: Complete set of human videos recorded in the *single-view* condition for each of the three tasks. Each task includes 20 videos captured from a fixed viewpoint.

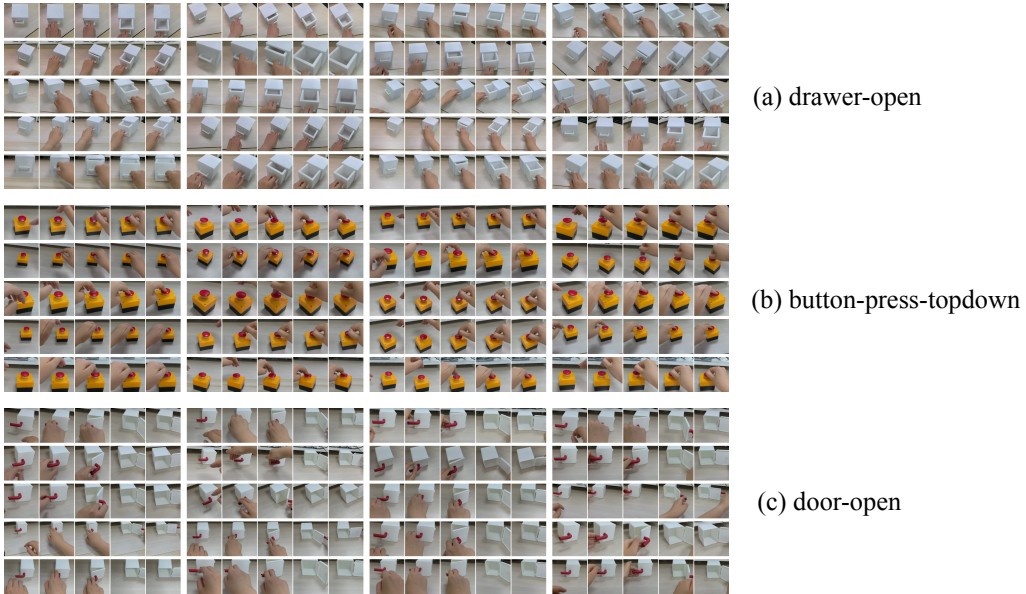

Figure 10: Complete set of human videos recorded in the *multi-view* condition for each of the three tasks. Each task includes 20 videos captured from varying viewpoints.

## A.3   ADDITIONAL EXPERIMENT RESULTS

### A.3.1   VIP BACKBONES

In the main experiments (Figure 5), we compare *TimeRewarder* with baseline methods including Rank2Reward and PROGRESSOR, both of which use the same ViT backbone as *TimeRewarder*.

However, VIP originally uses a ResNet34 backbone as recommended by its codebase. To ensure a fair comparison, we re-train VIP using the same ViT backbone as *TimeRewarder*.

As shown in Figure 11, the performance of VIP with ResNet34 versus ViT varies across different tasks. Sometimes, ResNet34 performs better, and other times, ViT shows superior performance. However, regardless of the backbone used, *TimeRewarder* consistently outperforms VIP across all tasks. This demonstrates that, while the choice of backbone can influence VIP's performance on individual tasks, *TimeRewarder* remains more robust and superior in all scenarios.

This setup follows the same experimental conditions as in Section 5.1, where all methods, including VIP, are evaluated on the same 10 Meta-World manipulation tasks using sparse binary success signals.

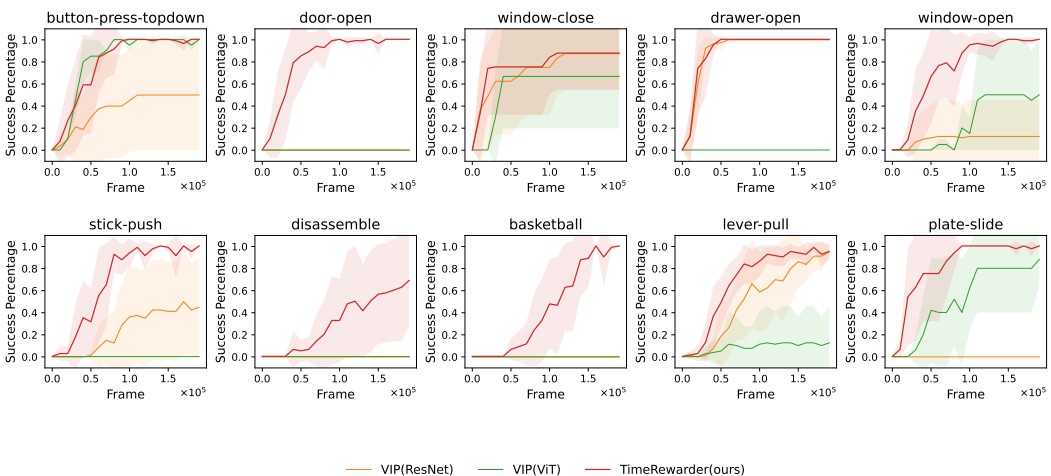

Figure 11: VIP performance with ResNet34 vs. ViT backbones across tasks. The results show that *TimeRewarder* outperforms VIP regardless of the backbone used. All methods are evaluated with reinforcement learning using sparse environment success signals and dense proxy rewards. Curves show the mean ± s.d. over eight seeds.

### A.3.2    RL WITH ONLY PROXY REWARD

Compared to Figure 5, Figure 12 presents the results when the environment's sparse reward is entirely removed, relying solely on the learned proxy reward. Additionally, we include results for the ILfO baseline **BCO** (Torabi et al., 2018a; Baker et al., 2022). Under the constraint of extremely short training (only 200,000 frames), no successes are achieved. However, by the end, the agent has started making progress and completing part of the task, though not the full goal.

### A.3.3    CHOICE OF REWARD COMBINATION FACTOR

In equation 7 we use a weight constant $\alpha$ to align the scales of the dense and sparse components so that neither term dominates purely due to magnitude differences, since different methods produce reward functions on different scales. In practice, we use an adaptive way to choose $\alpha$, which is to calculate the maximum value of the dense reward for the first few trajectories, and then set $\alpha$ to ten times this maximum value.

Figure 13 further shows that the performance of *TimeRewarder* is not sensitive to the value of $\alpha$.

### A.3.4    CHOICE OF DISCRETIZATION BINS

Figure 14 shows the performance of *TimeRewarder* with different bin number $K$ in 2-hot discretization. While the default $K = 20$ represents a practical tradeoff between reward precision and optimization stability, the performance is not very sensitive to the value of $K$.

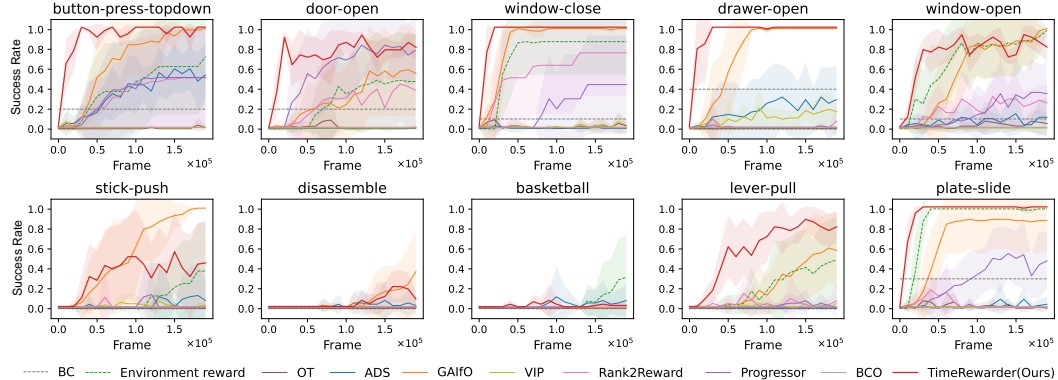

Figure 12: Reinforcement learning without sparse reward. Curves show mean ± s.d. over eight seeds. Dashed lines indicate behavior cloning (BC) and environment dense reward supervision.

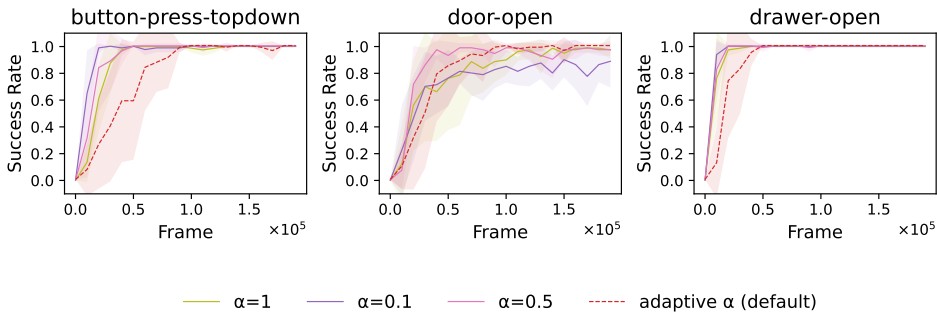

Figure 13: *TimeRewarder*'s performance with different $\alpha$. Curves show mean ± s.d. over eight seeds.

### A.3.5 ADDITIONAL CROSS-DOMAIN EXPERIMENTS

In the main experiment results, with 100 Meta-World demonstrations, performance is already strong. Adding 20 human videos yields similar performance but slightly improves sample efficiency by further broadening the coverage. The results are shown in Figure 15 .

### A.3.6 TRAINING FROM SCRATCH

Figure 16 presents an ablation study in which a ViT-B model trained from scratch is used as the visual backbone, replacing the CLIP-pretrained ViT-B employed in the main experiments. We compare TimeRewarder against the baseline methods Rank2Reward and PROGRESSOR. As expected, the absolute performance decreases due to the weaker visual representations. Nevertheless, the overall trends persist, and TimeRewarder continues to outperform the baseline methods.

### A.4 IMPLEMENTATION DETAILS

### A.4.1 ALTERNATIVE TEMPORAL MODELING APPROACHES

In our main method, *TimeRewarder* does temporal modeling through predicting the relative progress between two frames in a video. We also examined other three temporal modeling approaches as following.

**(1) only from init.** Considering the distribution shift, predicting the progress from each frame in an agent's rollout trajectory to a goal image derived from another expert trajectory may not be suitable. In addition to the goal frame, a natural choice is to use the initial frame as an anchor, which captures the positions of objects in the environment. In this context, when sampling frame pairs from expert trajectories, instead of randomly selecting any two frames, we fix the first frame as the initial frame.

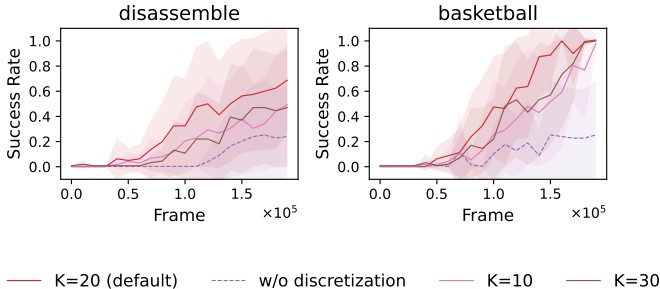

Figure 14: *TimeRewarder*'s performance with different bin number $K$ in 2-hot discretization. Curves show mean $\pm$ s.d. over eight seeds.

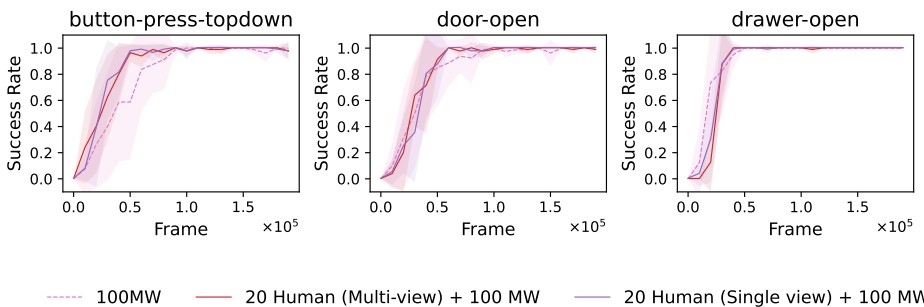

Figure 15: Cross-domain experiment results of adding 20 human videos to 100 Meta-World demonstrations. Curves show mean $\pm$ s.d. over eight seeds.

We then predict the progress within the range of [0, 1], while adhering to the three methodological components in *TimeRewarder*.

**(2) single frame input.** The simplest method to capture temporal information in a video is to directly predict the normalized temporal position (ranging from [0,1]) of each individual frame. In contrast to *TimeRewarder*, we use only one frame as input for our reward model instead of two. We uniformly sample the frame and apply the discretization technique.

**(3) order prediction.** Our order prediction setting is inpired by the setup of GVL (Ma et al., 2024). During training, we uniformly sample $n = 32$ frames from each expert video and apply a random permutation. The model is trained to recover the original ordering using a cross-entropy loss over permutation positions. At test time, we input an agent trajectory and predict a score for each frame reflecting its position in the estimated order. The model architecture mirrors that of *TimeRewarder*, but replaces the temporal regression head with a frame-wise classifier for permutation indices. Specifically, the predicted scalar values are normalized between $[-1, 1]$.

**Reward computation:** For all three methods mentioned above, the prediction of the reward model reflects the progress of an agent's trajectory at each time step. These scores are then utilized as potentials in a potential-based reward formulation. Consequently, the reward for each step is defined as the forward difference between successive predicted values.

### A.4.2 DEMONSTRATION COLLECTION FOR META-WORLD

To better approximate in-the-wild video data, we collected Meta-World demonstrations under a deliberately diverse initialization protocol. Rather than using the default narrow initialization range, where both agents and experts begin from nearly identical configurations, we expanded the initial state space to cover a broad variety of robot and object positions. This leads to demonstrations with much greater appearance diversity and prevents agent trajectories from being trivially aligned to demonstrations at the pixel level.

This choice also explains the results in Figure 5, where occupation-matching methods such as Optimal Transport (OT) and its extension ADS perform poorly. With the default narrow initialization,

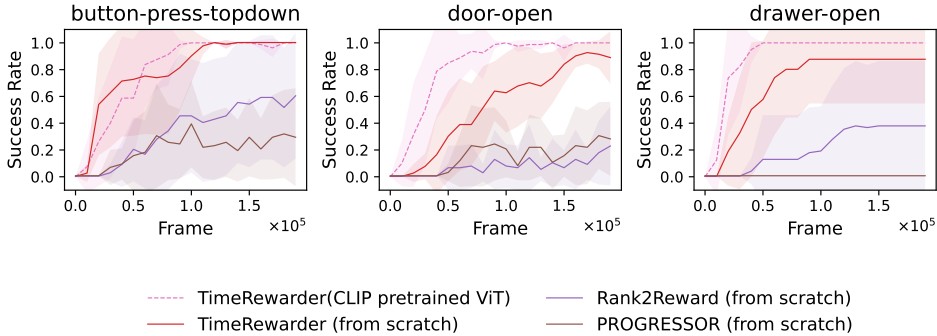

Figure 16: Ablation study with from-the-scratch ViT-B as the vision backbone. Curves show mean ± s.d. over eight seeds.

agents and experts share similar starting conditions, allowing OT and ADS to exploit appearance-level shortcuts when aligning trajectories. Once the initialization range is broadened, these shortcuts disappear, and the assumptions underpinning OT and ADS no longer hold, leading to degraded performance.

Crucially, this setting more faithfully reflects real-world conditions, where demonstrations and agent experiences seldom begin from the same initial states. It therefore underscores the importance of methods like *TimeRewarder* that extract robust progress signals rather than depending on superficial appearance matching.

### A.4.3 HYPERPARAMETERS

For reward learning, we use a ViT-B/16 backbone. Frame features are extracted, concatenated into a 1024-dimensional vector, and projected through a linear layer into 20 discretized bins. Training data is augmented to 10,000 pairs per epoch. The hyperparameters are summarized in Table 1.

Table 1: Reward model hyperparameters.

| Config | Value |
|---|---|
| Backbone | ViT-B/16 |
| Feature dimension | $1024 (512 \times 2)$ |
| Output bins | 20 (two-hot discretization) |
| Training pairs per epoch | 10,000 |
| Epochs | 100 |
| Warm-up epochs | 5 |
| Batch size | 16 |
| Accumulation steps | 1 |
| Optimizer | Adam |
| Learning rate | $2 \times 10^{-5}$ |

We equip all the methods with the same underlying RL algorithm, DrQ-v2 (Yarats et al., 2021). The hyperparameters are listed in Table 2.

Table 2: RL hyperparameters.

| Config | Value |
| --- | --- |
| Replay buffer capacity | 150000 |
| $n$-step returns | 3 |
| Mini-batch size | 512 |
| Discount | 0.99 |
| Optimizer | Adam |
| Learning rate | $10^{-4}$ |
| Critic Q-function soft-update rate $\tau$ | 0.005 |
| Hidden dimension | 1024 |
| Exploration noise | $\mathcal{N}(0, 0.4)$ |
| Policy noise | $\text{clip}(\mathcal{N}(0, 0.1), -0.3, 0.3)$ |
| Delayed policy update | 1 |

