# OpenReview forum: "TimeRewarder: Learning Dense Reward from Passive Videos via Frame-wise Temporal Distance"
_ICLR.cc/2026/Conference — Submitted to ICLR 2026_

### Official Review · Reviewer_igvB · 2025-10-25

**Soundness:** 2
**Presentation:** 3
**Contribution:** 2
**Rating:** 4
**Confidence:** 4

**Summary:**

The paper proposes a reward-learning method that turns passive videos (robot or human) into dense, step-wise rewards for RL. In detail, they learn a model to predict a frame-wise temporal distance between two observations. At RL time, the predicted temporal distance between adjacent frames serves as a progress reward, optionally combined with a sparse success signal. The authors emphasize three training choices: (i) implicit negatives via forward/backward ordering to encode regress vs. progress, (ii) exponentially weighted pair sampling to focus on short temporal gaps, and (iii) two‑hot discretization for stable training. For experiments, the method reportedly reaches near‑perfect success on 9/10 tasks with 200k interactions and, in most cases, outperforms both prior video‑based contenders and the environment’s dense reward on Meta‑World tasks.

**Strengths:**

- Simple objective with clear inductive bias. Framing reward learning as temporal distance prediction is intuitive and easy to implement.
- Thorough ablations on design choices. The study isolates the contributions of implicit negatives, weighted sampling, and two‑hot discretization. It is useful for practitioners.

**Weaknesses:**

- Originality is thinner than claimed. Temporal Distance Classification (TDC) was introduced in [1]; it discretizes time differences and trains a classifier on video frame pairs, which is very close to this paper’s core target. Related pretext tasks like Time‑Contrastive Networks and Temporal Cycle Consistency similarly extract progress‑like signals from temporal structure [2,3], and Shuffle&Learn / “arrow‑of‑time” also leverage ordering cues [4]. The paper cites TCN but does not cite TDC/TCC or clearly differentiate from them.
- Connections to recent progress‑from‑video rewards are not discussed. TimeRewarder is close in spirit to VIP (value‑implicit pretraining), GVL (VLMs as in‑context value estimators), Rank2Reward (temporal ranking for shaped reward), and HashReward (progress reward with online hashing). The paper positions against these but omits or soft‑pedals some important links and failure modes already discussed there (e.g., distribution shift and online refinement) [5–8].
- The paper gestures at potential‑based shaping $R(o_t, o_{t+1})=V(o_t) - \gamma V(o_{t+1}))$ but the learned signal is pairwise $F_\theta(o_t, o_{t+1})$, not an explicit potential $V(o). Without showing that $F_\theta$ ntegrates to a state potential (or at least is path‑independent), there is no policy‑invariance guarantee. In the absence of this, shaped rewards may induce loops or reward hacking in off‑expert states [9].

## Reference
[1] Playing Hard Exploration Games by Watching YouTube.

[2] Time-Contrastive Networks: Self-Supervised Learning from Video.

[3] Temporal Cycle-Consistency Learning.

[4] Learning and Using the Arrow of Time.

[5] VIP: Towards Universal Visual Reward and Representation via Value-Implicit Pre-Training.

[6] Vision Language Models are In-Context Value Learners.

[7] Rank2Reward: Learning Shaped Reward Functions from Passive Video.

[8] Imitation Learning from Pixel-Level Demonstrations by HashReward.

[9] Policy Invariance Under Reward Transformations: Theory and Application to Reward Shaping.

**Questions:**

Please refer to the weaknesses.

---

> ### Author Response · Authors · 2025-12-01
> **Response to Reviewer igvB**
>
> We thank Reviewer igvB for the detailed feedback. We address each of the concerns below.
>
> # W1: about relation to TDC and other works
> We thank the reviewer for highlighting additional related work, including TDC, TCC, and “arrow-of-time” methods, which we have incorporated into the Related Work section. As the reviewer notes, modeling temporal structure is an important way to learn from video in an unsupervised manner, and many prior works, including our own, explore this idea. Effectively capturing task progress from such signals, however, remains challenging.
>
> The core innovation of TimeRewarder is its direct, simple, and stable modeling of task progress: it predicts the temporal distance between any ordered frame pair, providing a dense and learnable signal that directly reflects task structure. While we use a 2-hot discretization, this is a practical trick to improve training stability and data efficiency, rather than as a core originality. This trick is fundamentally different from TDC, which treats temporal prediction as a multi-way classification over discrete bins. By encoding scalar temporal distances as sparse 2-hot vectors, TimeRewarder preserves continuity and fine-grained temporal structure, avoiding the information loss inherent in categorical labels and enabling more stable and effective learning.
>
> In addition, TimeRewarder differs significantly from TCC and “arrow-of-time” methods. TCC aligns two trajectories via a cyclic matching procedure rather than directly modeling task progress, while arrow-of-time approaches only predict the forward or backward direction of sequences, extracting very limited information and resembling rank-based signals like Rank2Reward. In contrast, TimeRewarder can extract richer temporal information from videos, allowing it to capture nuanced task progress that these prior methods cannot.
>
> # W2: about recent progress‑from‑video works
> We clarify that we have discussed the connections and limitations of related works, including VIP, GVL, and Rank2Reward, in the Related Work section. We also appreciate the pointer to HashReward, which we have now incorporated. We respectfully clarify that HashReward doesn’t explicitly model temporal progress but is essentially an extension of GAIL.
>
> Regarding distribution shift, all progress-based reward methods face trajectory-level out-of-distribution challenges because they learn to score non-expert rollouts from expert demonstrations. TimeRewarder differs in how it models progress: by predicting temporal distances between all ordered frame pairs, it captures both fine-grained transition-level information—including small correct or incorrect actions—and long-horizon task structure. This dense supervision fully exploits the information contained in each frame pair, providing generalization beyond what previous methods achieve. Our experimental results validate this advantage, and Figure 4 further visualizes how TimeRewarder provides more informative transition-level feedback than alternative methods.
>
> Regarding online refinement, while methods like Rank2Reward and PROGRESSOR note that offline-only training can limit performance, TimeRewarder demonstrates strong results purely from the offline dataset. This shows that its progress-based modeling is robust and highly informative without online updates. Importantly, our approach is orthogonal to online refinement and could be combined with it to further enhance reward quality if desired.
>
> # W3: about potential-based shaping
> It is true that we use a neural network to parameterize the potential difference, which does not guarantee that the resulting potential obtained by summation is path-independent. However, we would like to clarify that our training objective is mathematically derived from the potential itself, making it equivalent in form. As long as the network fits the potential difference well, it can approximate the potential closely enough. We acknowledge that this approach does not provide a formal policy-invariance guarantee and cannot fully rule out reward hacking. Nonetheless, in our experiments, we did not observe loops or unintended behaviors in off-expert states. We hypothesize that this is related to 3D physical constraints of the tasks themselves.

---

### Official Review · Reviewer_QyHg · 2025-10-26

**Soundness:** 3
**Presentation:** 4
**Contribution:** 2
**Rating:** 6
**Confidence:** 4

**Summary:**

Time rewarder presents a method of deriving a reward function from expert demonstrations through predicting task progress. Their reward model predicts the relative distance between two frames, normalized from -1 to 1, measuring both forward and negative progress. Time rewarder beats out chosen baselines, including both learning from observations only (OT ADS GAIfO) and other IRL methods (VIP Rank2Reward PROGRESSOR). They also include experiments that co-train their reward with human videos, and a provide a theoretical motivation for progress rewards.

**Strengths:**

This paper includes strong baseline comparisons across multiple tasks. They also included ablation studies explaining each of their design decisions, such as descritization, negative sampling, and relative vs. absolute progress. It is well written and well executed. The results on cross-embodiment reward transfer are also strong.

**Weaknesses:**

Using task progress or temporal ordering as a reward signal is not particularly novel. Several prior works (VIP, Rank2Reward, PROGRESSOR) already use temporal structure for reward learning. They note that other ranking-based methods are difficult to optimize reliably, and that rank2reward and PROGRESSOR report poor performance on out-of-distribution states. However, for their data used to train the ranking function, they collect demonstrations “under a deliberately diverse initialization protocol.”

Additionally, Time Rewarder uses a CLIP-pretrained ViT backbone. Prior work like RoboCLIP [1] has shown that a CLIP-pretrained representation can produce meaningful reward signals. It would help to isolate the contribution of the proposed method from that of the pretrained features. One way to do this would be a comparison using a non-contrastively pretrained backbone.

[1]  Sumedh Anand Sontakke, Jesse Zhang, Séb Arnold, et al. RoboCLIP: One Demonstration is Enough to Learn Robot Policies.

**Questions:**

* Given that the demonstration set is deliberately diverse, does the time rewarder require state coverage of expert demonstrations to ensure relative progress does not go out of distribution?  How does Time Rewarder perform under the standard expert demonstrations for Meta World?
* The behavior cloning policy seems to perform poorly. If the authors could explain why this occurs in their setting, it would help contrast with their method.
* What enables TimeRewarder to outperform dense-environment rewards?
* For co-training on the human videos, if you use both the human videos and the 100 meta-world demos, does this further improve performance compared to just the 100 demos?

---

> ### Author Response · Authors · 2025-12-01
> **Response to Reviewer QyHg**
>
> We thank Reviewer QyHg for their thoughtful review and valuable suggestions. The reviewer has raised several insightful points regarding the visual backbone’s role and the method’s assumptions, which are valuable for clarifying the scope and impact of our work. Below, we address each comment and question in detail.
>
> # Response to Weaknesses
> ## Part1: about task progress and previous works
> We appreciate the reviewer’s recognition that temporal progress is a strong supervisory signal. While the high-level idea is shared across prior work, effectively modeling and exploiting this signal remains challenging. Our approach offers a simple, effective, and scalable mechanism for leveraging temporal progress when learning rewards from videos.
>
> The limitations we highlight for Rank2Reward and PROGRESSOR are consistent with those reported in their original papers, including optimization difficulties and degraded performance on out-of-distribution states. Our choice to collect demonstrations under a deliberately diverse initialization protocol is driven by scalability: real-world human-collected or in-the-wild video datasets cannot rely on carefully controlled initial configurations. Designing the data this way ensures that the learned reward model must capture generalizable progress rather than memorize specific trajectories. This does not affect fairness: all methods, including ours and the baselines, are trained on the same dataset.
>
> ## Part2: about isolating the contribution of the visual backbone
> We thank the reviewer for raising this point. We agree that encoder capacity strongly influences representation quality. To isolate this factor from our method, in the revised paper, we add an experiment in **Appendix A.3.6** using a ViT trained from scratch. As expected, performance drops slightly, but the advantage of our progress-based reward formulation over baseline methods remains. Since TimeRewarder’s design is orthogonal to the encoder choice, stronger encoders naturally enhance reward quality without changing the core mechanism.
>
> # Response to Questions
> ## Q1: about state coverage
> Coverage of expert demonstrations is indeed important for TimeRewarder to generalize effectively and ensure that relative progress remains within distribution. This aligns with observations from large-scale LLM, VLM, and VLA pretraining, where broader coverage directly improves generalization. For this reason, all our baselines are trained on deliberately diverse demonstrations. TimeRewarder’s benefits of cross-domain human videos similarly arise from the increased coverage they provide.
>
> Under the standard MetaWorld demonstration setup, where expert and agent initializations are confined to a narrow range, reward-learning methods generally perform worse, while trajectory-matching approaches are less affected.
>
> In general, any method that aims to generalize must rely on either diverse data or carefully controlled conditions. As datasets scale up, coverage naturally increases, making the use of diverse demonstrations both reasonable and forward-looking.
>
> ## Q2: about BC performance
> Two factors contribute to this: (1) The number of demonstrations in MetaWorld is small. (2) behavior cloning (BC) suffers from compounding error and distributional shift. Consistent with other pixel-observation-based BC results reported on MetaWorld, like in the ROT paper [1], BC performance is generally weak in this setting.
>
> [1] Watch and Match: Supercharging Imitation with Regularized Optimal Transport
>
> ## Q3: about outperforming environment rewards
> TimeRewarder outperforms dense-environment rewards largely because the hand-crafted dense rewards in MetaWorld often fail to reflect pre-contact progress, which is crucial for effective control. In contrast, TimeRewarder captures even small, incremental advances seen in demonstration videos, offering a smoother and more informative learning signal than the manually designed reward shaping.
>
> ## Q4: about improving performance with human data
> With 100 MetaWorld demonstrations, performance is already strong. Adding 20 human videos yields similar performance but slightly improves sample efficiency by further broadening the coverage. We include the corresponding experiments in **Appendix. A.3.5** in the revised paper.

---

### Official Review · Reviewer_GauR · 2025-10-28

**Soundness:** 3
**Presentation:** 4
**Contribution:** 3
**Rating:** 6
**Confidence:** 4

**Summary:**

The paper proposes TimeRewarder, a method for learning dense rewards from action-free expert videos.
The key idea is to model task progress as a temporal distance prediction problem between frame pairs.
By self-supervisedly learning the temporal gap between two observations, the model implicitly captures how far the agent has progressed toward task completion.
The approach integrates three main components: implicit negative sampling, exponentially weighted pair sampling, and two-hot discretization.
Theoretically, the learned temporal distance corresponds to a potential-based reward-shaping term, ensuring Bellman consistency.
Experiments on ten Meta-World manipulation tasks show that TimeRewarder's rewards remain highly monotonic on unseen expert trajectories, successfully distinguish failed rollouts, and enable reinforcement learning agents to achieve top success rates.
Overall, TimeRewarder is simple, theoretically grounded, and empirically strong.

**Strengths:**

1. The paper presents a simple yet effective idea, treating reward learning as temporal-distance estimation, which provides a clean, self-supervised way to learn dense progress signals without requiring expert actions or environment rewards.

2. The connection between temporal distance and potential-based reward shaping gives the method solid theoretical justification and ensures consistency with RL principles.

3. Evaluations on ten Meta-World tasks are extensive, showing high sample efficiency and outperforming both prior progress-based methods and imitation baselines.

4. Through implicit negative sampling, the model can recognize and penalize regressive or failed behaviors, improving policy learning stability.

5. The ablation studies carefully isolate and validate the contribution of each design component, reinforcing the method’s architectural soundness.

**Weaknesses:**

1. The cross-domain generalization largely stems from the CLIP backbone's semantic representations rather than the proposed temporal-distance formulation itself. The method benefits from CLIP's object-centric and domain-invariant features, but this reliance makes it unclear how well TimeRewarder would perform with less powerful or task-specific encoders.
The paper does not analyze how different visual backbones or representations affect performance, making it hard to disentangle the contribution of TimeRewarder from that of pretrained features.

2. The core idea, learning task progress from temporal consistency, builds on prior progress-based reward learning work (e.g. VIP, PROGRESSOR, Rank2Reward) with incremental architectural refinements rather than a fundamentally new principle.

3. The method assumes that expert trajectories reflect smooth, monotonic advancement toward a goal. This assumption breaks in multi-stage or looping tasks where temporal distance no longer correlates with true progress.

4. The reward predictor is fixed during RL, which limits adaptability to out-of-distribution states encountered in exploration.

**Questions:**

1. In the human-to-robot transfer experiment, was one model trained per task, or a single model shared across all three tasks?

2. How much of TimeRewarder's generalization depends on the CLIP backbone? It would be helpful to include an ablation using non-pretrained or weaker encoders (e.g., a randomly initialized ViT or a task-specific CNN) to quantify the backbone's contribution.

3. The reward network is frozen during reinforcement learning. Would online fine-tuning or adaptive updating improve robustness to out-of-distribution states encountered during policy exploration?

---

> ### Author Response · Authors · 2025-12-01
> **Response to Reviewer GauR**
>
> We sincerely thank Reviewer GauR for the supportive and constructive review. We address all the concerns and questions below.
> # Response to Weaknesses
> ## W1: about the contribution orthogonal to the backbone
> We appreciate the reviewer’s observation. Indeed, TimeRewarder and the choice of visual backbone are orthogonal: a stronger visual encoder naturally improves representation quality, which in turn benefits any video-based reward model.
>
> To further clarify this point, in the revised paper, we have added experiments in **Appendix A.3.6** using a from-scratch ViT encoder. As expected, the performance is slightly poorer due to weaker representations, but the main trends and the advantages of TimeRewarder compared with baseline methods remain, confirming that the temporal-distance formulation, not only the backbone, is responsible for the observed behaviors.
>
> We agree that CLIP’s strong invariances help cross-domain performance; this is consistent with the broader literature; our focus is orthogonal to encoder design, and we believe future improvements in backbones can strengthen the method.
>
> ## W2: about progress-based reward learning
> We appreciate the reviewer’s recognition that progress is a powerful supervisory signal implicit in expert videos. This idea has indeed become a shared understanding in the community, and prior work such as VIP, PROGRESSOR, and Rank2Reward has begun to explore it. While we share the same high-level intuition about progress, our formulation and mechanism are straightforward and stable, offering a complementary, easy-to-use solution for future research on learning from videos.
>
> We believe that simplicity and stability are crucial. Many prior works, such as CLIP, have reduced algorithmic complexity compared to earlier approaches, improving robustness and reliability. In the same spirit, our straightforward formulation provides a stable and practical framework for learning from videos. We hope that this simplicity and robustness will make it easier for the community to adopt, extend, and scale progress-based reward learning in future research.
>
> ## W3: about multi-stage or looping tasks
> We fully agree with the reviewer’s observation. As noted in our limitations section, progress alignment becomes more challenging in multi-stage or looping tasks. This limitation is inherent to all video-based progress models.
>
> Hierarchical or memory-augmented progress modeling is a promising direction. Importantly, the atomic element in such hierarchical systems would still rely on a unit like TimeRewarder to measure progress within subsegments.
>
> ## W4: about online refinement
> We appreciate the reviewer’s comment. Online refinement of the reward predictor using rollout data is indeed a meaningful future direction, and we plan to explore it in follow-up work.
>
> In this paper, we focus on the pretraining stage: building a reward model that generalizes well without requiring online updates. Our aim is to understand how far strong pretraining alone can go, and our experiments show that this approach is effective even without online refinement. Such refinement can be incorporated at any time, and we expect it to become even more beneficial as the pretraining scale increases in future work.
>
> # Response to Questions
> ## Q1: about human-to-robot transfer experiment
> We trained separate TimeRewarder models per task in the main experiments. In principle, a single TimeRewarder model could also handle multiple tasks simply by conditioning on text descriptions.
>
> ## Q2: about ablation using non-pretrained encoders
> As noted in W1, backbone strength and the temporal-distance formulation are conceptually orthogonal. We have added a randomly initialized ViT experiment in the revised paper to illustrate this. With any future improvements in vision encoders, TimeRewarder will naturally benefit, but its core mechanism remains unchanged.
>
> ## Q3: about online fine-tuning
> As mentioned in W4: yes, it is possible, and we agree it may help in certain OOD cases.

---

### Official Review · Reviewer_X7My · 2025-10-29

**Soundness:** 1
**Presentation:** 1
**Contribution:** 2
**Rating:** 2
**Confidence:** 4

**Summary:**

The paper proposes TimeRewarder, a method that learns a dense, step-wise reward for reinforcement learning directly from passive expert videos by predicting the temporal distance between pairs of frames. The authors propose learning a model that outputs per-step rewards along an agent’s trajectory that signal progress in successful completion of the task, combined with weighted sparse task reward. To stabilize their proposed method, the use the following:
1. *Implicit Negative Sampling*: The word sampling is misleading here. They mean to say that their temporal distance reward can be positive or negative depending on the order of the frames in the recorded trajectory
2. *Exponentially Weighted Pair Sampling*: Sample adjacent frames more frequently than distant frames
3. *Two-hot Discretization*: An auxiliary loss to stabilize training; this seems to stem from an intuition that classification losses are more numerically stable than regression losses in deep learning.

The evaluate their results in simulation (MetaWorld) and compare with some relevant competitors.

**Strengths:**

1. Clear motivation and problem setting
  - Addresses the practical challenge of dense reward design for manipulation from only passive videos, without action annotations.
  - Connects progress-based shaping to a directly learnable quantity—temporal distance—which is readily available from videos.
2. Simple idea to benefit from human demonstrations provided using videos

**Weaknesses:**

**Writing quality and clarity**
The paper’s presentation can be improved. Several sections, particularly the method description and experimental setup, lack clarity. The idea can be stated in a far simpler way than the current form.

**Overstated empirical claims**
The paper overstates the efficacy of the proposed method. Reported improvements are not statistically significant in several plots, and standard errors overlap in almost all cases. The tone implies strong superior performance, but the evidence supports minimal gain at best.

**Limited applicability beyond simple goal-reaching tasks**
The method appears well suited for short-horizon goal-reaching tasks  (go from point A to point B). It is unclear how it would perform in environments with loops, subgoals, or frequent revisits to the same state. Because the reward is based on framewise temporal distance, states with high visitation counts is likely to confuse the model and degrade performance.

**Weak cross-domain evidence**
The claim of strong cross-domain generalization is not sufficiently supported. The real human video experiments appear tightly controlled: object positions and scene layouts closely match the simulation, with only the robot hand replaced by a human one. This suggests limited diversity and minimal domain shift. Additional experiments with different backgrounds, object appearances, and camera viewpoints would be needed to substantiate the cross-domain claim.

**Questions:**

1. **Statistical significance of results:**
   In most figures (for example, Figures 3, 5, 6b, and 7), the standard error bands of different methods overlap substantially. On what basis do you claim that TimeRewarder performs better? The results as shown do not appear statistically significant.

2. **Generalization to out-of-distribution trajectories:**
   How does TimeRewarder handle out-of-distribution data? With only a few demonstrations, it seems unlikely that the model can generalize to every possible trajectory. In Meta-World, what happens if the robot begins from a different initial configuration or follows a novel trajectory? How would this affect the predicted reward? Would this be a fundamental limitation of the approach?

3. **Weighting of the success reward:**
   Why is a separate weighting factor for the success reward needed in Equation 7? How sensitive is performance to the choice of $\alpha$, and how difficult is it to tune in practice?

4. **Choice of discretization bins:**
   The two-hot discretization uses (K = 20) bins. How was this number selected, and how sensitive are the results to this value?

5. **Transfer to real robots:**
   All reported experiments appear to be conducted in simulation. How do you expect the method to extend to real-world robotic systems, given practical challenges such as lighting variation, occlusion, and partial observability?

---

> ### Author Response · Authors · 2025-12-01
> **Response to Reviewer X7My - Part1**
>
> We sincerely thank Reviewer X7My for their time and insightful comments. In this response, we address each concern point-by-point.
>
> # Response to Weaknesses
>
> ## W1: Writing quality and clarity
>
> We appreciate the reviewer's feedback regarding the clarity of certain sections. As an aside, it's worth noting that all three other reviewers explicitly gave us positive ratings for our presentation. To address this, we will revise the organization and simplify the explanations to enhance clarity in the camera-ready version.
>
> ## W2: Overstated empirical claims
>
> We appreciate the reviewer’s concern regarding the statistical interpretation of our results. We would like to clarify that the shaded regions in the plots represent the population standard deviation of returns, not the standard error of the estimator. This shading reflects the inherent variability of returns in reinforcement learning tasks, where different runs can naturally diverge due to exploration randomness, environment stochasticity, and the sensitivity of policy learning. High across-run variance is typical in RL benchmarks and does not itself indicate weak methodological effects.
>
> Each data point in our figures is computed from 8 independent runs. The relevant estimator uncertainty (standard error of the mean) is therefore the population standard deviation divided by √8, which is substantially smaller than the shaded regions. Thus, the visual overlap of the shaded areas should not be interpreted as evidence that performance differences lack statistical significance; these areas depict variability in the underlying return distribution rather than uncertainty in the estimated mean.
>
> We hope this clarification helps prevent misinterpretation of the plotted statistics and provides a clearer view of the empirical results.
>
> ## W3: Limited applicability beyond simple goal-reaching tasks
> Thank the reviewer for highlighting the limitation. At its core, the issue arises from partial observability: short-horizon visual observations alone do not provide enough information to disambiguate revisited states, and a reward defined solely through frame-wise temporal distance cannot resolve such ambiguities.
>
> Crucially, this challenge is fundamentally about state representation rather than the objective our method aims to address. Complementary approaches for mitigating partial observability, such as memory-augmented VLA architectures, can be integrated with our framework. Incorporating structured memory or recurrent state encoders would likely address these challenges and extend the method to more complex tasks.
>
> In our work, we assume full observability, as our focus is on the basic challenge of low-level manipulation, where fine-grained visual states are essential but still difficult to model. Under this fully-observable design, our method achieves strong performance, and future extensions incorporating richer state representations offer a promising direction.
>
> ## W4: Weak cross-domain evidence
>
> We acknowledge the concern regarding the domain shift between the real human video and simulator. However, it is important to emphasize that the real and simulated domains exhibit significant differences beyond the mere replacement of a robot hand with a human one. Specifically, there are notable disparities in texture, lighting conditions, and camera viewpoints, as the human videos are captured using handheld mobile phones, resulting in substantial variability in perspective and environmental factors.
>
> In contrast to many real-world robotic setups, which rely on fixed, calibrated cameras, our approach with handheld devices provides a more scalable and dynamic setup. Additionally, we conduct experiments under multi-view conditions, where the camera perspectives are varied to further amplify the domain shift. These multi-view experiments account for significant shifts in appearance and spatial configuration, all of which present nontrivial challenges for cross-domain generalization.
>
> While our current experiments demonstrate TimeRewarder's potential to leverage cross-domain video data, we recognize the need for more diverse datasets. Future work will focus on expanding to larger, in-the-wild datasets to further validate and improve cross-domain generalization.

---

> ### Author Response · Authors · 2025-12-01
> **Response to Reviewer X7My - Part2**
>
> # Response to questions
> ## Q1: Statistical significance of results
> As discussed above in W2, the overlap of the shaded areas does not indicate a lack of statistical significance across methods, as they represent variability in the underlying return distribution, not estimator uncertainty.
>
> ## Q2: Generalization to out-of-distribution trajectories
> Thank the reviewer for pointing out the central challenge addressed in this work: reward generalization from action-free demonstrations to previously unseen trajectories. Although we only use a small number of demonstrations, each provides many supervision signals because every ordered frame pair contributes a temporal-distance learning target. Consequently, even for some out-of-distribution trajectories, there can be transition-level similarities to the training data that TimeRewarder can exploit.
>
> Our experiments already evaluate this generalization ability in both the VOC setting (Figure 3) and downstream RL tasks (Figures 4–6), where the agent encounters novel initial configurations and trajectories not present in the demonstrations. In VOC, held-out expert demonstrations differ in both initial conditions and motion styles, ensuring exposure to unseen scenarios. In downstream RL, the agent’s exploration naturally leads it into states and action sequences that diverge significantly from any expert trajectory.
>
> We acknowledge that, like all methods trained from limited demonstrations, performance will degrade on trajectories that deviate too far from the training distribution. However, this limitation is not inherent to our approach. TimeRewarder is orthogonal to the dataset and backbone choices, and it already outperforms relevant baselines. With larger-scale in-the-wild data and stronger general-purpose vision backbones, we expect the generalization challenge to be further alleviated.
>
> ## Q3: Weighting of the success reward
> The weighting factor is needed because different methods produce reward functions on different scales, while the sparse success reward is fixed at 0/1. The factor in Equation 7 aligns the scales of the dense and sparse components so that neither term dominates purely due to magnitude differences.
>
> We additionally include experiments in **Appendix A.3.3** in the revised paper, showing that the method is not sensitive to the choice of $\alpha$. In practice, finding a suitable value for this parameter is straightforward. We use an adaptive way to choose α for all methods, which is to calculate the maximum value of the dense reward for the first few trajectories, and then set $\alpha$ to ten times this maximum value.
>
> ## Q4: Choice of discretization bins
> Our ablation in Figure 7 shows that even without the two-hot discretization, performance remains strong on most tasks—specifically, 8 out of 10 manipulation tasks are largely unaffected. Selecting K=20 represents a practical tradeoff between reward precision and optimization stability. To further assess sensitivity, we include an additional experiment in **Appendix A.3.4** in the revised paper, which demonstrates that the method is not highly sensitive to the choice of K. Using K = 10 or K = 30 also yields improvements over the no-discretization baseline.
>
> ## Q5: Transfer to real robots
> Real-world RL faces challenges such as safety, sample efficiency, and hardware limitations. Our method is highly scalable and can leverage videos without action annotations, with approximate or noisy demonstrations, and captured under diverse conditions. Combined with its cross-domain generalization ability, this enables the use of large in-the-wild or easily recorded human videos, supporting future real-robot learning and providing a practical path toward deployment.

---

### Author Response · Authors · 2025-12-03
**Summary to Reviews and Responses**

We appreciate the encouraging comments and acknowledgement of the value of our work from all reviewers. Especially, GauR and QyHg highlighted the simplicity, stability, and strong empirical performance of TimeRewarder, including its thorough ablations and cross-embodiment results. X7My noted the clear problem motivation, and the usefulness of learning progress from passive videos without actions. igvB emphasized the strong inductive bias and the careful isolation of each design choice.

## Additional Experiments Added in the Rebuttal
In our rebuttal revision, we added the following experiments in the Appendix:
1. **Visual encoder ablation (GauR, QyHg).**
We added a ViT-from-scratch variant, showing that the performance gains persist even with a weaker backbone. This supports that the benefits come from our temporal-distance formulation itself and are not dependent on the choice of visual backbone.
2. **Analysis of $\alpha$ and $K$ (X7My).**
We extended robustness analysis across the weighting factor $\alpha$ and discretization bin count $K$ and found both hyperparameters to be insensitive within reasonable ranges and do not materially affect conclusions.
3. **Further human video co-training (QyHg).**
We further evaluated human-video co-training, showing that adding a small amount of human video data to already diverse demonstrations yields modest but consistent improvements in sample efficiency without changing overall trends.


## Clarifications to Other Concerns

1. X7My interpreted overlapping shaded bands as evidence of weak statistical significance. We clarified that the shading reflects population variability in RL returns rather than uncertainty of the reported mean return across 8 seeds. In support of this, other reviewers such as GauR and QyHg explicitly recognized the empirical results as strong and well-supported, citing the extensive baselines, ablations, and consistent task performance.

2. Regarding revisited states and looping tasks (X7My, GauR), we clarified that observed limitations arise from partial observability rather than the objective itself and can be addressed with memory-augmented VLA architectures.

3. On domain shift (X7My), we clarified that human videos differ from simulator data in texture, lighting, and viewpoint, amplified by handheld and multi-view capture. Despite these substantial shifts, TimeRewarder still benefits from such supervision.

4. For igvB, we clarified distinctions from related work such as TDC and HashReward and explained how our potential-based formulation approximates path-independent shaping in practice without observed reward hacking.

Overall, reviewers agreed on the clarity of the motivation, the simplicity and stability of the temporal-distance formulation, and the strength of the empirical evidence. These experiments and clarifications directly address the main concerns raised by the reviewers.

---

### Meta-Review · Area_Chair_kbqL · 2026-01-12

**Summary:**

The paper proposes a framework that leverages passive expert videos to predict the temporal distance between pairs of frames, which serves as a proxy for dense reward modeling. The authors show the superiority of the method compared to previous baselines across multiple Meta-World tasks.

During the rebuttal, the authors have provided additional experiments and clarifications that addressed multiple concerns. However, several concerns remain regarding the method's robustness and scope. Additionally, the novelty of the work seems quite limited. Given the high expectation bar for ICLR, these factors make it difficult to recommend the acceptance of the paper.

**Reviewer Concerns:**

The paper has mixed opinions with two borderline accept (6) and one borderline reject (4) and one reject (2).

Following major concerns were addressed by the rebuttal.

Orthogonality to the CLIP backbone (Reviewer `GauR`, `QyHg`); the authors address this in Appendix A.3.6, and show that their method still outperforms baselines when using ViT from scratch, despite minor performance degradation.

Limited Empirical Gains (Reviewer `X7My`); the authors clarify that standard error of the mean is substantially smaller, thereby supporting the superiority of their method.

Revisited states and looping  (Reviewer `X7My`, `GauR`): the authors point out that such limitations arise from partial observability in the state representation, reasonably justifying this issue as beyond the scope of the current paper.

Outstanding Concerns

Limited novelty (Reviewer `GauR`, `QyHg`, `igvB`): The authors acknowledge that modeling temporal progress has been many times studied in previous works. Also, while scalability is suggested, it is not rigorously tested beyond a single simulation domain. In this light, concerns regarding novelty remain despite the author’s rebuttal.

Limited Cross-domain evidence (Reviewer `X7My`):  Despite the disparities such as texture, lighting conditions, and camera viewpoints, the fact that object positions and scene layouts closely match the simulation weakens the cross-domain generalization argument.

(partially resolved) Generalization and State Coverage (Reviewer `X7My`, `QyHg`): While the authors state that OOD generalization is evaluated via VOC and RL, their reliance on data collection with “a deliberately diverse initialization protocol” brings certain concern. This is particularly notable given that separate models are trained for each individual task, which allows the diverse data to easily saturate the narrow state space of a single task. While the authors justify this strategy with an analogy to scaling, the claim remains questionable without empirical evidence on how the method behaves in different domains with more complex state spaces.

**Reviewer Scores:**

All reviewers are likely to maintain their original score.

---

### Decision · Program_Chairs · 2026-01-26

Reject